# Corporate Social Responsibility: The Impact of Employees' Perceptions on Organizational Citizenship Behavior through Organizational Identification

**Carla Freire * , Joana Gonçalves and Maria Rita Carvalho**

School of Economics and Management, University of Minho, 4710-057 Braga, Portugal
* Correspondence: cfreire@eeg.uminho.pt

**Abstract:** The impact of the employees' perceptions of CSR activities on the employee-organization relationship has been little explored in the literature. This study aims to analyze the impact of corporate social responsibility (CSR) perceptions (Environment-related CSR; Employee-related CSR, Customer-related CSR) on organizational citizenship behavior through the mediating role of the organizational identification of employees in the hotel sector. A questionnaire was applied to a sample of 214 frontline employees who work in four- and five-star hotels. Using Model 4 of the PROCESS macro, a mediation model was designed to test the hypotheses. The results indicate that there is a mediation model that uses the effect of organizational identification in the relationship between perceptions of CSR and organizational citizenship behavior. This study concludes that an increase in identification with the organization based on environmental-related activities, employee-related activities, or customer-related activities impact on OCB. The results of this study represent contributions to the literature in the CSR field, as well as to the case of human resources managers who wish to enhance organizational citizenship behavior and identification among hotel staff.

**Keywords:** CSR perceptions; environment-related CSR; employee-related CSR; customer-related CSR; organizational identification; organizational citizenship behavior; hotels

## 1. Introduction

The employee-organization relationship is influenced by employees' perceptions of CSR activities (Oh et al. 2021; George et al. 2021), namely, the employees' perspective on how impactful the social responsibility activities developed by the company are to its beneficiaries and society (Bhattacharya et al. 2009; Kim et al. 2010; Park and Levy 2014; Song et al. 2019).

Companies that develop more CSR practices are considered to be more attractive, since socially responsible activities contribute to the creation of a favorable image and reputation, and are thus able to attract the best employees (e.g., Porter and Kramer 2011). This is particularly true of younger candidates who tend to value environmentally friendly corporate policies (PwC—PricewaterhouseCoopers 2011). In this sense, the literature "acknowledges the value-driven role of CSR" (Malik 2015, p. 419); subsequently, companies tend to launch efforts to become good organizational citizens through various activities and programs designed to benefit multiple stakeholders (e.g., the environment and community, employees and customers). Therefore, when employees have a positive perception of the company's values and CSR initiatives, they tend to identify more closely with the company, and are more predisposed to exert extra efforts to achieve organizational objectives.

The literature has shown that CSR activities developed by companies positively influence employees' attitudes and behaviors (e.g., Clarkson 1995; Kim et al. 2010), which especially impacts on organizational citizenship behaviors (e.g., Zhao and Zhou 2019) and on employees' sense of identification with the organization (e.g., Park and Levy 2014).

Organizational citizenship behavior (OCC) is defined as a set of discretionary behaviors in the workplace that exceed basic job requirements and are often described as behaviors that go beyond the call of duty (Smith et al. 1983). For Organ (1988), OCBs are discretionary behaviors–not directly or explicitly recognized by the formal reward system– and which, as a whole, promote the effective functioning of the organization (Organ 1988). Such behaviors include voluntary acts of creativity and innovation designed to improve the organization's task or performance (Podsakoff et al. 2000). In other words, it can be understood as discretionary individual behavior that is advantageous to the organization (van Dick et al. 2008). Due to its obvious advantages for the organization, OCB has become an appealing construct, both for academics as well as practitioners, insofar as it focuses on aspects of work that lead the individual to attempt to go the extra mile in order to favor the organization. Accordingly, several studies have sought to understand what makes individuals want to be good organizational citizens (Coyle-Shapiro et al. 2004). One explanation focuses on the logic of reciprocity, in which employees engage in this type of behavior in order to reciprocate for the way the organization acts and treats them. Blau's (1964) theory, based on the logic of reciprocity, can serve as a theoretical framework to explain the influence of organizational policies on the behavior of organizational members, constituting a good starting point to explain this link. The reasons underlying this need experienced by individuals to give back to the organization are associated with a sense of identification.

Organizational identification has been considered by the literature to be "a significant driver of group members' attitudinal and behavioral support" (Park and Levy 2014, p. 336), reflecting the quality of employee-organization relationships (Bhattacharya et al. 2009). Consequently, Ashforth et al. (2008) have posed important research questions, such as: "How does identification occur?"; more specifically, "How do individuals bring attributes of the organization's identity into their own identities?"; and also, "Why does identification matter?"

Faced with the question of what factors lead to identification with the organization, the response provided by the literature has been diverse, pointing to "self-enhancement" and "self-consistency" (Ashforth et al. 2008). Above all, responses reveal that individuals basically seek to establish bonds with others, in the sense that they feel they belong to a group (Ashforth and Mael 1989; Ashforth et al. 2008). The answer to the last of the aforementioned questions seems to be that organizational identification (OI) matters when one considers individual outcomes: according to the authors, it "helps capture the essence of who people are and, thus, why they do what they do—it is at the core of why people join organizations and why they voluntarily leave, why they approach their work the way they do and why they interact with others the way they do during that work. Identification matters because it is the process by which people come to define themselves, communicate that definition to others, and use that definition to navigate their lives, work-wise or other" (Ashforth et al. 2008, p. 334). However, despite individual outcomes, the literature has also pointed to an extensive list of relevant positive organizational outcomes, such as job satisfaction and work adjustment (e.g., Carmeli et al. 2007), as well as the intention to stay and absenteeism, extra-role behavior, work involvement (e.g., Carmeli et al. 2007; Riketta 2005), and organizational citizenship behaviors (e.g., van Dick et al. 2006; Freire and Pieta 2022). Accordingly, studies indicate that if there is identification with the organization, workers show more favorable attitudes and behaviors toward the organization and seek to increase their efforts in order to respond positively to the needs of the organization. Within this context, one is then able to understand why identification matters for organizations— it is due to the fact that, when employees "wear the company t-shirt," they are more predisposed to making sacrifices and giving more for its sake, thus developing extra-role efforts toward the organization (Freire and Pieta 2022).

Despite the importance of OI, few studies have explored its mediating role in explaining how employees perceive CSR activities and how they manifest a predisposition to develop OCBs. Based on these previous considerations, this study aims to explore the

role of OI as a mediating mechanism in the relationship between perceptions of CSR and OCB. Accordingly, this study aims to contribute to the ongoing reflection on CSR regarding the importance of the employee´s perceptions of CSR practices, and how it impacts on positive behaviors toward the organization. Thus, this study seeks to provide insights into the mediating mechanisms through which CSR perceptions—based on the environment, employees, or the customer—influence OCB.

This study begins with a review of the literature on corporate social responsibility activities and then explores the relationship between perceptions of CSR on OCB and perceptions of CSR on OI, to anchor the study's hypotheses. The following section presents the methods, including the study context, measures used, procedures, and sample. The following section presents the results obtained in the statistical analysis. The discussion of the results obtained follows, and finally, the conclusions of the study are presented.

## 2. Literature Review

### 2.1. Corporate Social Responsibility Activities

The definition of the CSR concept has evolved over decades of research (e.g., Rodriguez-Gomez et al. 2020) and is associated with various objectives for companies. These include promoting its reputation, company image and products/services, as well as value creation through stakeholders (Fernández-Guadaño and Sarria-Pedroza 2018), or even as a fundamental element of the company's sustainability strategy (Dey et al. 2018). Investment in socially responsible initiatives can contribute to a closer relationship between employees and organizations (Sen et al. 2006; Kim et al. 2010) in different ways: when socially responsible activities contribute to the well-being of employees; when measures are taken to protect the environment (Su and Swanson 2019); or when companies are actively involved in their community (Peterson 2004; Helming et al. 2016; Gursoy et al. 2019; Su and Swanson 2019).

Individuals who are more aware of the importance of CSR present more positive associations with the organization (Sen et al. 2006; Gaudencio et al. 2020). Seen as a long-term strategy (Rhou and Singal 2020), the adoption of social responsibility practices is an investment that promotes stakeholders' satisfaction, additionally providing greater commitment to organizations (Gursoy et al. 2019). Social responsibility initiatives are rather diverse, and include activities such as: collaborating in social projects in the local area; adapting facilities to accommodate people with disabilities; implementing good team practices in terms of equality and non-discrimination; developing training programs; and encouraging customers to consume local products (Suárez-Cebador et al. 2018). Environmental impact is reflected, for example, in the choice of ecological suppliers and ecological products, and communicating this practice to the customer. Socially responsible management can generate countless benefits when well implemented: it guarantees sustainability; provides a good organizational climate; and adds value to the company's image and products, as well as to the brand itself. Though less explored but equally beneficial is the issue of the impact and perception of these responsible management practices on employees' behavior and attitudes, namely in the creation of bonds and their identification with the organization, as well as on OCB.

### 2.2. Perceptions of Corporate Social Responsibility and Organizational Citizenship Behavior

Understanding the relationship between the perception of social responsibility and OCB is an issue that the literature has recently begun to dedicate itself to with greater determination. In order to analyze this relationship, the theories of social identity (Tajfel 1974) and social exchange (Blau 1964) are particularly relevant as they enable a better understanding of the influence of the perception of corporate social responsibility practices on employees' positive behavior. The results of the study by Jones (2010) revealed that employees reciprocate through organizational citizenship behavior when they perceive socially responsible business practices. Using Blau's (1964) theory of social exchange, more specifically the notion of reciprocity, one can consider that organizational members may perceive that social responsibility is also in their interest, as the main stakeholders of

organizations, seeking to repay the organization's investments in practices of this nature (Glavas 2016). This exchange is sustained by reciprocity: when organizations guarantee good working conditions and benefits for their employees, the latter, in turn, reciprocate through extra efforts (Story and Neves 2015; Glavas 2016). Similarly, Hansen et al. (2011) also use social exchange theory (Blau 1964) to explain why employees engage in discretionary behavior when they wish to "repay" their organization for its benevolence toward them, or in exchange for the organization's commitment or positive relationship with them. According to this theory, when employees perceive that their organization adopts socially responsible practices, they are more likely to exert extra effort in performing their duties and are less likely to consider leaving the company (Hansen et al. 2011).

Perceptions of social responsibility and organizational citizenship behavior are more pronounced in employees with high levels of moral identity (Rupp et al. 2013). From this perspective, employees who share the same ethical values adopted by the organization generally show more organizational citizenship behaviors (Baker et al. 2006). For example, the perceptions of organizational support in relation to the environment are significantly related to the attitudes and behaviors of employees, namely through behaviors that go beyond what is stipulated in favor of sustainability (Lamm et al. 2015). Noteworthy is another form of response to social responsibility initiatives by employees; that is, discretionary CSR actions, such as community donations, which do not provide direct benefits to employees; however, they, as members of the community (and of society in general), tend to reciprocate these activities in the form of positive organizational behaviors (Kim et al. 2017). That is, when employees perceive that their organization invests in social responsibility practices, either intrinsically or extrinsically (when seeking to obtain some benefit), they tend to exercise extra effort in their work (Story and Neves 2015).

Thus, when employees perceive the organization to be socially responsible, they will tend to reciprocate and develop OCBs. As such, hypothesis 1 was elaborated on:

**Hypothesis 1 (H1).** *Frontline employees' perceptions regarding social responsibility practices (H1a: Environment-related CSR; H1b: Employee-related CSR; H1c: Customer-related CSR) will be positively related to OCB.*

*2.3. Perceptions of Corporate Social Responsibility and Organizational Identification*

Organizational identification is a concept that is commonly defined and based on a relationship that the individual establishes with the organization. Studies have resorted to Tajfel's theory of social identity (1974); this is grounded in the importance of belonging to certain social groups, and the emotional and evaluative meaning of that belonging. According to Ashforth and Mael (1989), the concept of organizational identity is based on this notion of belonging, but extends to the existence of a "oneness with" or "unity" to an organization, whereby the experience of the organization's successes and failures are experienced as if they were the individual's own. Or simply stated: organizational identification is a kind of "psychological attachment" that occurs "when members adopt the defining characteristics of the organization as defining characteristics of themselves" (Dutton et al. 1994, p. 242). From this perspective, when organizational identification is strong, it means that there is an incorporation of what is "distinctive, central, and enduring" in the organization (Dutton et al. 1994, p. 242). Therefore, people tend to truly identify with the organization when their identity as an organizational member is more important than other identities, and concept of self possesses many of the same characteristics that define the organization as a social group (Dutton et al. 1994).

OI has been defined in different ways in the literature, thus pointing to broader or narrower conceptualizations (e.g., Edwards 2005; Dutton et al. 1994; Dutton et al. 2010; Ashforth and Mael 1989; Ashforth et al. 2008). For example, in a broader perspective, Ashforth and Mael (1989, p. 21) define the concept as "the perception of oneness or belongingness to some human aggregate". In this same broad approach, Dutton et al. (1994, p. 239) define it as the moment when "a person's self-concept contains the same attributes

as those in the perceived organizational identity". This theoretical reference in the literature is important insofar as it allows one to understand why certain members of organizations show a greater sense of identification than others.

Similarly, the model proposed by Rousseau (1998) also enables an understanding of why people identify with the organization and proposes two types of identification: situational and deep structure. The first, situational identification, is more likely to occur when individuals and the organization have common interests, and the results of actions can be shared by both. This type of identification is expected to end as soon as the task is completed and can be formed rather quickly. In this context, it is related to the success of temporary jobs, and may prove relevant to new forms of work. When situational identification occurs, the individual's interaction with the organization may become deeper over time, giving rise to deep structure identification. The identification of deep structure occurs when what is experienced by the individual in the organization is significant. This then leads to the formation of mental models, which are capable of exerting influence on the way individuals come to see themselves. As organizations are subjected to many alterations, the second form of identification may encounter barriers due to the constant changes in values, objectives, and incentive systems present in contemporary organizations (Rousseau 1998).

Individuals' identification is also influenced by others' perceptions of the organization (Dutton et al. 1994). The model presented by the previous authors focuses on two key images, namely those of perceived organizational identity and the interpreted organizational image obtained from external members. According to the authors, these images "shape the strength of members' identification with the organization, serving as important cognitive reference points that connect or disconnect a member of the organization" (Dutton et al. 1994, p. 258). The first is related to the members' evaluation of the organization's character ("members' beliefs about the organization's distinctive, central, and enduring attributes"), while the second concerns members' beliefs as to how outsiders evaluate the organization. An example of this assumption is when employees have a sense of how customers (external stakeholders) perceive their organization (Glavas and Godwin 2013).

From these ties that members establish with the organization they are a part of, several relevant consequences can ensue, both for individuals as well as the organizations involved (Ashforth and Mael 1989). In this sense, organizations must pay special attention to the factors that can affect and determine them. For this purpose, the literature has explored various antecedents of the organizational identification construct. Ashforth and Mael (1989) consider that attributes such as personal interaction, similarity, preferences, proximity, shared goals and a common history increase the tendency for an individual to identify with the organization. Added to this background are its unique characteristics, more specifically what differentiates it from the others; its prestige; and the salience of other organizations (Ashforth and Mael 1989). Other antecedents of identification studied include leadership (Ehrhart 2004; Walumbwa et al. 2011), perceived organizational support (Sluss et al. 2008; Subba 2019), and social responsibility (He and Brown 2013).

Regarding the last antecedent mentioned, Farooq et al. (2014) used Tajfel's theory of social identity (1974) to explain that social responsibility measures promote organizational identification, both in the case of internal stakeholders and external parties (e.g., community and customers), as well as good deeds on the part of employees, due to the measures implemented. Identification with the organization can be strengthened by the importance employees have attributed to social responsibility activities (Glavas and Godwin 2013). In this context, this study aims to examine how employees perceive CSR practices and how these practices (centered on the environment, customers, or employees) influence their identification with the organization.

Based on this, we formulated the second hypothesis:

**Hypothesis 2 (H2).** *Frontline employees' perceptions regarding social responsibility practices (H2a: Environment-related CSR; H2b: Employee-related CSR; H2c: Customer-related CSR) will be positively related to OI.*

Employees are increasingly looking for organizations whose concerns are not restricted to the business itself. Namely, they wish to be part of companies whose concerns go beyond what is required, thus seeking companies that establish ties and good relationships with other stakeholders and the environment (Su and Swanson 2019). The truth is that social responsibility practices contribute to an organization's positive image, improving its reputation and prestige and, eventually, distinguishing it from other organizations (Ehrhart 2004; Carmeli et al. 2007; Doh et al. 2011; Park and Levy 2014), in this case in the hotel sector. From this perspective, organizations in the tourism sector that show responsible behavior ultimately benefit their reputation, thus improving their image with external stakeholders, as well as creating a favorable climate of trust and cooperation within the company (Camilleri 2016). Accordingly, social responsibility policies and practices are expected to serve as an example for the model of behavior valued by the organization and, consequently, influence the manifestation of identification and subsequent organizational citizenship behaviors.

As stakeholders, employees form perceptions regarding internal social responsibility practices (or initiatives related to social responsibility aimed at the environment, the employees and customers) (De Roeck and Maon 2018). Employees' perceptions of the company's social responsibility practices positively influence their attitudes and behaviors, namely organizational citizenship behavior (Rupp et al. 2006, Slack et al. 2014). Thus, according to the norm of reciprocity, if the organization exhibits social responsibility practices and if these practices benefit the environment, the individual or the customers; the employees will, in turn, feel a greater sense of identification with the organization and reciprocate through behaviors that are beneficial to the organization, such as OCB. As such, the third hypothesis was formulated:

**Hypothesis 3 (H3).** *Frontline employees' perceptions regarding social responsibility practices (H3a: Environment-related CSR; H3b: Employee-related CSR; H3c: Costumer-related CSR) will be positively related to organizational citizenship behaviors through the mediating role of organizational identification.*

## 3. Method

This section presents the research method. Accordingly, the context, measures, procedure, and sample are presented as follows.

### 3.1. Research Context

Considered as one of the main industries of the 21st century (Aynalem et al. 2016), the hotel sector is gaining a real awareness of the significance of sustainability and, more specifically, the importance given to employees and their perception of this type of practice (Holcomb and Smith 2015). Despite existing studies, most research on this topic has focused on the company and customers' perspectives (Kim et al. 2018). The analysis of employees' perceptions of social responsibility activities is extremely pertinent for the management of the hotel industry since its employees, who are in direct contact with customers, play a decisive role in the success of these organizations (Kim et al. 2018).

The tourism sector is characterized by a high level of turnover, poorly paid work, job insecurity, seasonality, unpromising professional careers, and few development opportunities, among other aspects (Chytiri et al. 2018). In this context, typical features of the sector are the often-irregular working hours, which are usually poorly paid. In addition, these are subject to pressures inherent to the functions performed which, in many circumstances, can lead to burnout (Mansour and Tremblay 2016). Due to the unstable, transitory, and low-

skilled working conditions, there is a high turnover of staff, which represents unavoidable costs for these organizations (DiPietro and Bufquin 2018).

Investment in socially responsible initiatives by hotels can also be associated with the adoption of environmentally and socially responsible practices in the workplace (Su and Swanson 2019), such as: adapting facilities to accommodate people with disabilities; implementing good practices in terms of equal opportunities and non-discrimination; developing training programs; encouraging customers to consume local products (Suárez-Cebador et al. 2018) and restricting their use of natural resources, like water and electricity (Molina-Azorín et al. 2009); or even establishing a relationship with the local inhabitants (Gursoy et al. 2019), such as collaboration in social projects within the community (Rhou and Singal 2020).

*3.2. Measures*

In order to measure CSR perceptions, 22 items from Park and Levy (2014) were used, more specifically to assess employees' perceptions of socially responsible practices in the hotel industry. The original authors divided the scale into three dimensions: CSR-Environment and Community (eleven items), CSR-collaborators (six items), and CSR-Customers (five items). Some examples of the items are: "My hotel encourages guests to reduce their environmental impact through programs and initiatives" (CSR-Environment and Community), "My hotel encourages guests to reduce their environmental impact through programs and initiatives" (CSR-employees). "My hotel treats our employees fairly and respectfully," and "Customer satisfaction is very important to my hotel" (CSR-Customers). For each item, respondents were asked to indicate the degree of agreement using a Likert-scale consisting of five levels of agreement (where 1 corresponds to "Strongly Disagree" and 5 to "Strongly Agree").

In order to measure organizational identification, the 6-item organizational identification scale by Mael and Ashforth (1992) was used. Bearing in mind that organizational identification is the perception of unity or belonging to the organization (Mael and Ashforth 1992), this scale aims to assess the degree to which respondents identify with the organization they are a part of. Some of the items considered are: "When someone criticizes the hotel, I feel it as a personal insult," "I am really interested in what others think about the hotel," "The success of this hotel is my success." Using the study carried out by Lu et al. (2016) as a reference, some items were adapted, due to the context in which the study was carried out, so that the term "organization" was replaced by the term "hotel".

In order to measure organizational citizenship behavior (OCB) in relation to the organization, 5 items of the scale devised by Ma et al. (2013) were used. This organizational dimension of the scale describes the citizenship behaviors adopted by employees regarding the organization. Some examples are: 'I will give advance notice if I cannot come to work,' and "I protect our hotel's property." A 5-point Likert-scale was used to assess the respondent's level of agreement regarding organizational citizenship behaviors, based on the following response possibilities: 1, 'Strongly Disagree'; 2, 'Disagree'; 3, 'Do not agree or disagree'; 4, 'Agree' and 5, 'Strongly Agree'.

*3.3. Procedures*

In order to test the questionnaire after translating the scales into Portuguese and adapting it to the context of the study, the questionnaire application phase was preceded by a pre-test with 17 employees of three-star hotels. At this stage, whenever possible, we sought to directly question respondents about any doubts, criticisms, or suggestions to be introduced in the data collection instrument. Overall, the questionnaire proved to be well structured and understandable. Regarding the size of the questionnaire, it was considered unnecessary to reduce the number of items, especially since the average response time was around 10 min. After this stage, the final questionnaire pertaining to this study was applied. As the scale refers to the perception of social responsibility depending on the respondent's interaction with the organization's customers, the questionnaires were

answered by employees of the hotel sector who are in direct contact with customers. Prior to the application of the questionnaire, a formal collaboration request was prepared and sent via email to each of the hotels. Due to the difficulty in obtaining authorization from the Board, face-to-face contact was required so as to ensure a high number of responses. At this stage of delivery and collection of the analysis instrument, one had to access the site to obtain the answers and renew/remind respondents of the request to fill in the questionnaires.

The choice of providing a hard copy of the questionnaire ensued from the difficulty in contacting responsible staff via e-mail and obtaining answers, as well as the suggestion by some hotels that indicated a preference for this methodology. Despite this option, one also had to create a digital version of the questionnaire, disclosing the answer link in requests placed via e-mail. In total, six questionnaires were completed online, and the rest were returned on paper.

The introductory part of the questionnaire presented the study objective, and respondents were clearly informed as to the guarantees of confidentiality and anonymity of their answers. In this way, we sought to reinforce that this was a scientific study that had been authorized by the hotel's management, and we further appealed to the participants' sincere responses. Informed consent was obtained from the study participants. The email address and contact information were provided in the case of further queries about the study. After collecting data, a previous inspection allowed for the exclusion of some incomplete questionnaires, which would not be subjected to statistical treatment.

### 3.4. Sample

Questionnaires were obtained from 214 employees of the hotel sector. The study was applied to employees of four-star and five-star hotels in the North of Portugal. For the collection and calculation of the sample, the National Tourism Register (RNT) was used. The application of the questionnaires took place in the first three months of 2020. The implementation of social responsibility practices is considered to be increasingly relevant in this sector, namely with regard to the issue of compliance with sustainability requirements, so that hotels can be classified as four- and five-star accommodations. Questionnaires were received from four-star (n = 169, 79%) and five-star hotels (n = 45, 21%), from respondents aged over 17 and under 60, corresponding to an average of 32 years of age (Table 1). Concerning the gender of the respondents involved: of these 114 (53.5%) were female and 99 (46.5%) were male. Regarding the professional activity performed, 119 (55.6%) worked in the reception area, 68 (31.8%) in bars and restaurants, 9 (4.2%) worked in the cleaning and maintenance of facilities, 8 (3.7%) were involved in sports and recreational activities, 7 (3.3%) performed functions such as accommodation technician, operational management, group coordination, and head of the contracting department. In relation to the work schedule, 6.5% work part-time and 93.5% have full-time jobs. As for academic qualifications, participants are ranked as follows: 1 (0.5%) completed the 1st cycle of basic education, 1 (0.5%) completed the 2nd cycle of basic education, 13 (6.1%) completed the 3rd cycle of basic education, 79 (36.9%) completed secondary education, 12 (5.6%) possess a Bachelor's degree, 87 (40.7%) have an Honors' degree, and 20 (9.3%) have a Master's/Postgraduate degree.

Regarding seniority in the organization reported by the respondents, the answers indicated that about 50% of them had worked in the organization for less than 1 year, and the rest had worked for longer; of these, 6.6% had been with the organization for more than 10 years.

**Table 1.** Sample characterization.

| | N (%) |
|---|---|
| Gender | |
| Female | 114 (53.5%) |
| Male | 99 (46.5%) |
| Age | |
| 17–30 | 151 (74%) |
| 31–40 | 34 (16.7%) |
| 41–50 | 13 (6.4%) |
| 51–60 | 6 (2.9%) |
| Academic qualifications | |
| 1st cycle of basic education | 1 (0.5%) |
| 2nd cycle of basic education | 1 (0.5%) |
| 3rd cycle of basic education | 13 (6.1%) |
| Secondary education | 79 (36.9%) |
| Bachelor's degree | 12 (5.6%) |
| Honors' degree | 87 (40.7%) |
| Master's/Postgraduate degree | 20 (9.3%) |
| Professional activity | |
| Reception area | 119 (55.6%) |
| Bars and restaurants | 68 (31.8%) |
| Cleaning and maintenance | 9 (4.2%) |
| Sports and recreational activities | 8 (3.7%) |
| Accommodation technician, operational management, etc. | 7 (3.3%) |
| Work schedule | |
| Part-time | 14 (6.5) |
| Full-time | 200 (93.5) |
| Seniority in the organization | |
| <1 year | 97 (45.5) |
| 1–5 years | 82 (38.5) |
| 5–10 years | 20 (9.4) |
| >10 years | 14 (6.6) |

## 4. Results

The scales that made up the questionnaire were submitted to an exploratory factor analysis of the principal components in order to use the resulting factor scores in subsequent statistical analyses. In this phase of the analysis, two of its main assumptions were considered; namely, the value obtained in the KMO test (Kaiser-Meyer-Olkin) and Bartlett sphericity test, both of which evaluate the feasibility of carrying out factor analysis. According to recommendations in the literature, for each scale, items with loadings greater than 0.50 were selected for each factor (Howell 1992). Components with internal consistency coefficients greater than 0.70 were the only ones considered from the selected components (Nunnally 1978).

Based on these assumptions, exploratory factor analysis was then undertaken, as can be seen in Table 2. The highest scores correspond to a higher level of perception of corporate responsibility practices (environment/community, employees, and hotel customers), organizational identification, and organizational citizenship behaviors.

**Table 2.** Exploratory factor analysis.

| Variables | Items | Factor Loading |
|---|---|---|
| Environment-related CSR | "My hotel reports the hotel's environmental performance." | 0.866 |
| | "My hotel incorporates environmental concerns in its business decisions." | 0.840 |
| | "My hotel buys products and services locally, which minimizes environmental impact." | 0.831 |
| | "My hotel actively tries to reduce the environmental impact of its activities." | 0.821 |
| | "My hotel encourages guests to reduce their environmental impact through programs and initiatives." | 0.818 |
| | "My hotel helps to improve the quality of life of the local community." | 0.790 |
| | "My hotel encourages employees to become involved with community organizations." | 0.769 |
| | "My hotel financially supports the environmental initiatives of other organizations." | 0.760 |
| | "My hotel incorporates community interests in its business decisions." | 0.740 |
| | "My hotel actively works with national/international organizations that promote responsible business." | 0.720 |
| | "My hotel financially supports local charities through donations, wind sponsorships, and/or provides goods and services." | 0.660 |
| | Explained variance (%) | 61.7 |
| | Cronbach's Alpha | 0.937 |
| Employee-related CSR | "My hotel's policies promote a good work-life balance for employees." | 0.867 |
| | "My hotel incorporates employees' interests in its business decisions." | 0.845 |
| | "My hotel encourages employees to develop their skills and careers," | 0.830 |
| | "My hotel offers a safe and healthy working environment for all employees," | 0.823 |
| | "My hotel treats our employees fairly and respectfully." | 0.798 |
| | "My hotel offers fair and reasonable wages." | 0.721 |
| | Explained variance (%) | 66.5 |
| | Cronbach's Alpha | 0.895 |
| Customer-related CSR | "One of my hotel's main principles is to offer high-quality services and products to our customers." | 0.840 |
| | "My hotel incorporates customer interests in its business decisions." | 0.834 |
| | "My hotel is sensitive to our customers' complaints." | 0.829 |
| | "My hotel respects consumer rights beyond legal requirements." | 0.774 |
| | "Customer satisfaction is very important to my hotel." | 0.753 |
| | Explained variance (%) | 65.09 |
| | Cronbach's Alpha | 0.863 |

**Table 2.** *Cont.*

| Variables | Items | Factor Loading |
|---|---|---|
| Organizational Identification | "I am genuinely interested in what others think about the hotel." | 0.796 |
| | "The success of this hotel is my success.". | 0.765 |
| | "When someone praises my hotel, I feel it to be a personal compliment." | 0.743 |
| | "If a media story criticized the hotel, I would be embarrassed." | 0.733 |
| | "When I talk about this hotel, I say "us" more often than "them."" | 0.709 |
| | "When someone criticizes the hotel, I feel it to be a personal insult." | 0.683 |
| | Explained variance (%) | 54.60 |
| | Cronbach's Alpha | 0.822 |
| Organizational citizenship behaviors— Organization | "I say good things about our hotel when talking to outsiders." | 0.712 |
| | "I protect our hotel's property." | 0.705 |
| | "I actively promote the hotel's products and services." | 0.688 |
| | "I follow informal rules in order to maintain order in the hotel." | 0.574 |
| | "I will give advance notice if I cannot come to work." | 0.561 |
| | Explained variance (%) | 32.88 |
| | Cronbach's Alpha | 0.950 |

All the scale items had loadings greater than 0.50, so none were excluded from the analysis (Nunnally 1978); the internal consistency values for all the scales and the explained variance were also above what is recommended in the literature (Nunnally 1978).

The correlational matrix (Table 3) presents the mean values, standard deviation, and correlation coefficients of the study variables. The average values indicate that employees consider social responsibility practices to be directed mainly at the hotel's customers. Practices which target the protection of the environment, or working conditions, are less valued if one considers the mean value. The results of the correlations indicate that organizational identification is positively correlated with the Environment-related CSR variable (r = 0.303; $p < 0.001$), Employee-related CSR variable (r = 0.260; $p < 0.001$), and Customer-related CSR (r = 0.269; $p < 0.001$). The results also indicate that OCB is positively correlated with the Environment-related CSR variable (r = 0.307; $p < 0.001$), Employee-related CSR variable (r = 0.205; $p < 0.001$), Customer-related CSR variable (r = 0.227; $p < 0.001$) and OI (r = 0.513; $p < 0.001$). This suggests that, when employees in the hotel sector perceive CSR practices, they will tend to develop a sense of greater identification with the organization they work for and are, subsequently, likely to develop organizational citizenship behaviors. These results will be explored in the next step, which deals with the estimation of subsequent regression models.

**Table 3.** Means, standard deviations and correlations of construct-related variables.

| Variable | Mean | STD.DEV. | 1 | 2 | 3 | 4 | 5 |
|---|---|---|---|---|---|---|---|
| Environment-related CSR | 3.405 | 0.874 | (0.937) | | | | |
| Employee-related CSR | 3.673 | 0.847 | 0.589 *** | (0.895) | | | |
| Customer-related CSR | 4.428 | 0.595 | 0.581 *** | 0.591 *** | (0.863) | | |
| OI | 3.972 | 0.710 | 0.303 *** | 0.260 *** | 0.269 *** | (0.822) | |
| OCB | 4.330 | 0.435 | 0.307 *** | 0.205 *** | 0.227 *** | 0.513 *** | (0.950) |

Note: N = 214 *** Significant at $p < 0.001$ (2-tailed). Cronbach alpha (appear along the diagonal in italics).

In order to test the hypotheses formulated and consider the effects of mediation, a bootstrapping method was performed using model 4 of the Process Macro (Hayes 2017), thus allowing for the analysis of the direct and imdirect effects of a single mediator. The purpose of this analysis is to verify if OI mediated the relationship between perceptions of CSR (Environment-related CSR; Employee-related CSR; Customer-related CSR) and OCB. Firstly, regression testing for mediation was conducted to verify whether OI mediated the relationship between Environment-related CSR and OCB. The regression of Environment-related CSR (X) on OCB (Y) was significant ($\beta = 0.1926$, t(214) = 3.2660; $p < 0.01$), thus confirming H1a (Table 4). The results of the regression analysis also indicated that Environment-related CSR (as an independent variable) was a significant predictor of OI ($\beta = 0.2214$, t(214) = 3.3331; $p < 0.01$), thus confirming H2a. Results of the regression analysis showed that, when controlling for OI (mediator), Environment-related CSR was significant as a predictor of OCB ($\beta = 0.5077$, t(214) = 8.3492; $p < 0.001$). The mediator, OI, accounted for approximately 57% of the total effect on OCB. Based on 5000 bootstrap samples, the results of the indirect effect showed a significant indirect relationship between Environment-related CSR and OCB (a*b = 0.1124, Bootstrap $CI_{95}$ = 0.0406 and 0.2019), thus allowing one to confirm the mediation effect of OI on the relationship between the perception of CSR based on the environment and OCB (confirming H3a). In sum, the mediation analysis revealed that OI partially explains the effect of Environment-related CSR on OCB; in addition, Environment-related CSR influences OCB, regardless of the proposed mediation mechanism ($\beta = 0.1926$, $p < 0.01$). Given that the direct effect is smaller than the total effect (Baron and Kenny 1986), one can infer that there is complementary partial mediation (Zhao et al. 2010).

**Table 4.** Regression Coefficients, Standard Errors, and Model Summary Information (Mediation Model_Environment-related CSR).

| Variable/Effect | β | SE | t-Value | CI 95% (LL-UL) |
|---|---|---|---|---|
| Environment-related CSR→OCB | 0.1926 | 0.0590 | 3.2660 ** | 0.0763–0.3089 |
| Employee-related CSR→OI | 0.2214 | 0.0664 | 3.3331 ** | 0.0904–0.3524 |
| Environment-related CSR→OI→OCB | 0.5077 | 0.0608 | 8.3492 *** | 0.3878–0.6276 |
| Direct effect | 0.1926 | 0.0590 | 3.2660 ** | 0.0763–0.3089 |
| Indirect effect | 0.1124 | 0.0406 | | 0.0406–0.2019 |
| Total effect | 0.3050 | 0.0665 | 4.5891 *** | 0.1740–0.4361 |

Notes: Based on a bootstrap test (5.000 re-samples). When the bootstrap of 95% CI (LL: lower levels; UL: upper level (JS)s) contains zero for one of the values. it indicates that the effect was not significant. β = Regression Coefficients; SE = Standard Error; CI = Confidence Interval; ** $p < 0.01$; *** $p < 0.001$.

Secondly, regression testing for mediation was conducted to verify whether OI mediated the relationship between the perception of CSR based on employees' practices and OCB. The regression of Employee-related CSR (X) on OCB (Y) was significant ($\beta = 0.1180$, t(214) = 1.9963; $p < 0.05$), thus confirming H1b (Table 5). The results of the regression analysis showed that Employee-related CSR (as an independent variable) was a significant predictor of OI ($\beta = 0.1602$, t(214) = 2.3831; $p < 0.05$), thus confirming H2b. While controlling for OI (mediator), Employee-related CSR was significant as a predictor of OCB ($\beta = 0.5329$, t(214) = 8.7363; $p < 0.001$). The mediator, OI, accounted for approximately 30% of the total effect on OCB. The results of the indirect effect pointed to a significant indirect relationship between Employee-related CSR and OCB (a*b = 0.0854, Bootstrap CI95 = 0.0012 and 0.1835), thus confirming H3b, i.e., the mediation effect of OI on the relationship between the perception of CSR focusing on the employee and OCB. Based on the same assumptions for the mediation mechanism (Baron and Kenny 1986; Zhao et al. 2010), the analysis showed that OI partially mediates the effect of Employee-related CSR on OCB ($\beta = 0.1180$, $p < 0.05$).

**Table 5.** Regression Coefficients, Standard Errors, and Model Summary Information (Mediation Model_Employee-related CSR).

| Variable/Effect | β | SE | t-Value | CI 95% (LL-UL) |
|---|---|---|---|---|
| Employee-related CSR→OCB | 0.1180 | 0.0591 | 1.9963 * | 0.0014–0.2345 |
| Employee-related CSR→OI | 0.1602 | 0.0672 | 2.3831 * | 0.0277–0.2927 |
| Employee-related CSR→OI→OCB | 0.5329 | 0.0610 | 8.7363 *** | 0.4126–0.6531 |
| Direct effect | 0.1180 | 0.0591 | 1.9963 * | 0.0014–0.2345 |
| Indirect effect | 0.0854 | 0.0461 | | 0.0012–0.1835 |
| Total effect | 0.2033 | 0.0683 | 2.9777 ** | 0.0687–0.3380 |

Notes: Based on a bootstrap test (5.000 re-samples). When the bootstrap of 95% CI (LL: lower levels; UL: upper level (JS)s) contains zero for one of the values. it indicates that the effect was not significant. β = Regression Coefficients; SE = Standard Error; CI = Confidence Interval; * $p < 0.05$; ** $p < 0.01$; *** $p < 0.001$.

In order to test the mediating role of OI in the relationship between hotel employees' perceptions of internal social responsibility practices regarding customers and OCB, a third model was estimated (Table 6). Results obtained indicated that the regression of Customer-related CSR (X) on OCB (Y) was significant (β = 0.1328, t(214) = 2.2346; $p < 0.05$), thus confirming H1c. The results showed that Customer-related CSR was a significant predictor of OI (β = 0.1775, t(214) = 2.6346; $p < 0.05$) (confirming H2c). Controlling for OI (as a mediator), Customer-related CSR was significant as a predictor of OCB (β = 0.5281, t(214)= 8.6539; $p < 0.001$). The mediator, OI, accounted for approximately 31% of the total effect on OCB. The results of the indirect effect pointed to a significant indirect relationship between Customer-related CSR and OCB (a*b = 0.0938, Bootstrap CI95 = 0.0002 and 0.1814), thus confirming H3c. Results indicated that OI partially mediates the relationship between the perception of CSR based on the customer and OCB (β = 0.1328, $p < 0.05$).

**Table 6.** Regression Coefficients, Standard Errors, and Model Summary Information (Mediation Model_Customer-related CSR).

| Variable/Effect | β | SE | t-Value | CI 95% (LL-UL) |
|---|---|---|---|---|
| Customer-related CSR→OCB | 0.1328 | 0.0594 | 20.2346 * | 0.0156–0.2501 |
| Customer-related CSR→OI | 0.1775 | 0.0674 | 20.6346 ** | 0.0447–0.3104 |
| Customer-related CSR→OI→OCB | 0.5281 | 0.0610 | 80.6539 *** | 0.4078–0.6485 |
| Direct effect | 0.1328 | 0.0594 | 20.2346 * | 0.0156–0.2501 |
| Indirect effect | 0.0938 | 0.0460 | | 0.0002–0.1814 |
| Total effect | 0.2266 | 0.0683 | 30.3172 ** | 0.0919–0.3613 |

Notes: Based on a bootstrap test (5.000 re-samples). When the bootstrap of 95% CI (LL: lower levels; UL: upper level (JS)s) contains zero for one of the values. it indicates that the effect was not significant. β = Regression Coefficients; SE = Standard Error; CI = Confidence Interval; * $p < 0.05$; ** $p < 0.01$; *** $p < 0.001$.

## 5. Discussion

In this study, we explored how employees perceive the socially responsible activities of their organizations, how these perceptions influence their level of organizational identification and, consequently, how they behave as good organizational citizens. In order to analyze these relationships, a model was tested, which uses the mediating role of OI in the relationship between CSR perceptions and OCB.

The answers obtained in this research constitute theoretical and empirical contributors to the analysis of the quality of employee-organization relationships (Bhattacharya et al. 2009; Lee et al. 2012). Accordingly, and in order to further our knowledge on this topic, which was mainly explored in the hotel sector (e.g., Camilleri 2016; Gursoy et al. 2019)

with frontline employees (e.g., Park and Levy 2014; Freire and Gonçalves 2021), this study concludes that frontline employees' perceptions regarding internal social responsibility practices (based on the environment, employees and customer) influenced OCB, and that this relationship was mediated by OI.

The results of this study indicated that, despite the fact that frontline employees consider hotel CSR practices to be directed mainly at the hotel's customers, the truth is that all the dimensions (environment, employees, and customers) contribute to the staff's identification with the organization (confirming H1a, b and c and H2a, b and c). As such, OI contributes to the development of OCB (confirming H3a, b and c).

This study expands on the existing literature (e.g., Park and Levy 2014; Oh et al. 2021; George et al. 2021) and seeks to contribute to the study of CSR as a company activity aimed at various stakeholders, including employees and customers, as well as the environment and the community (Postel and Sobel 2019). Considering the purpose of CSR–that companies repay society responsibly for what they receive from it–then, in line with this logic, and as internal stakeholders, employees "are essential for the existence and sustainability of the business" (George et al. 2021, p. 1094). The results indicated that hotel CSR practices directed at employees, the environment, or the customers, contribute to their "wearing the organizational t-shirt." In this sense, hotels must improve their CSR-related practices and communication with employees if they are to enhance the quality of the employee-organization relationship.

The results of this study pointed out that CSR practices are created in the hotel sector to address the interests of clients, the environment, and employees. Employees' CSR perceptions influence the staff's level of organizational identification, ultimately resulting in citizenship behavior. These results reveal employees' altruistic values (Chen and Choi 2008) because, when they feel they identify with the organization's CSR practices, they tend to reveal a predisposition toward defending the organization's interests and go further in the performance of their duties.

In sum, and in line with other studies in the literature, the results of this research indicate that managers are likely to "go green" (Park and Levy 2014). However, the results obtained in this study showed that organizations should implement CSR practices for their employees due to their role as "primary internal stakeholders" (Bhattacharya et al. 2009; Park and Levy 2014; George et al. 2021). In this sense, those in management must improve communication, and implement ways of listening to employees, if they wish to understand what staff expect in terms of participation and social intervention, what they consider to be a priority, and which stakeholders should be involved. In this regard, De Roeck and Maon (2018) emphasize the idea that company leaders should consider their employees to be a link between internal CSR activity and the external environment. As such, staff members should be involved in the sustainability strategies designed by the company and, in this sense, the training of employees must be provided to ensure they will become the company's "ambassadors" abroad. Furthermore, CSR activities contribute to the creation of a positive organizational image, which has been considered to be a determining aspect of employees' pride and willingness to belong to the organization (De Roeck and Delobbe 2012). In addition, employees should also contribute to the definition of a communication policy and digital marketing strategy for the CSR activities developed by the company.

We believe to have contributed to the advancement of knowledge in this field, mainly by presenting empirical evidence regarding the mediation mechanism between the perceptions of CSR and OI in the hospitality sector. However, future studies should seek to include other sectors, namely education or even industry and services. Despite these contributions, we have found some limitations in the study. First of all, the sample may not fully represent the opinion of frontline hotel employees. Secondly, the procedures followed in the application of the questionnaires (on paper and/or in digital format) may have somehow influenced the effect of the social desirability of the responses obtained. Future studies should thus seek to implement different forms of the application of the questionnaires to counter this potential effect. Moreover, future analyses should seek to

include other variables in the model in order to extend it and, for example, detect variables that may have an additional mediating effect on the relationship between the perceptions of CSR and OCB.

## 6. Conclusions

In short, the empirical evidence presented in this study shows that an increase in identification with the organization based on environmental-related activities, employee-related activities, or customer-related activities, impact on OCB. As such, employees in the hospitality sectors studied perceived socially responsible practices which, subsequently contributed to the enhancement of their "attachment" to the organization. In this context, they are more predisposed to going the extra mile in favor of the organization. To this end, organizations should focus their intervention on CSR practices if they wish to potentiate a sense of attachment to the organization, as well as the enhancement of citizenship behaviors.

**Author Contributions:** Conceptualization, C.F., J.G. and M.R.C.; methodology, C.F., J.G. and M.R.C.; software, C.F., J.G. and M.R.C.; validation, C.F., J.G. and M.R.C.; formal analysis, J.G. and M.R.C.; writing—original draft preparation, C.F.; writing—review and editing, C.F., J.G. and M.R.C.; supervision, C.F. All authors have read and agreed to the published version of the manuscript.

**Funding:** This research received no external funding.

**Institutional Review Board Statement:** Ethical review and approval were waived for this study because anonymity, privacy, and confidentiality were guaranteed to the survey participants.

**Informed Consent Statement:** Informed consent was obtained from all subjects involved in the study.

**Data Availability Statement:** The data will be made available on request from the corresponding author, Carla Freire, cfreire@eeg.uminho.pt.

**Conflicts of Interest:** The authors declare no conflict of interest.

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
