# Peer review of "Corporate Social Responsibility: The Impact of Employees’ Perceptions on Organizational Citizenship Behavior through Organizational Identification"

_admsci, doi:10.3390/admsci12030120_

Round 1
Reviewer 1 Report (Previous Reviewer 1)
I would like apologize for my english because i know it is not perfect.
First, i would like start by thanking the authors for the effort made in the review work.
Although they have substantially improved the article, they still need to improve it by not addressing some of the comments indicated above.
There is no research data sheet where it is clear which is the universe, the sample, the moment of time in which it is carried out, the confidence level, etc. This should be specifically indicated in the sample.
Author Response
Dear reviewer,
We are very grateful for the constructive comments. In reviewing the article, we took into account the aspects highlighted and we believe that the suggestions contributed to improve the article. In this sense, the section of the method was changed in order to present in more detail the sample. We thank again all the suggestions made.
Reviewer 2 Report (Previous Reviewer 3)
1. I recommend structuring the abstract as follows:
· Background: Place the question addressed in a broad context and highlight the purpose of the study; Methods: briefly describe the main methods or treatments applied;
· Results: summarize the article main findings;
· Conclusions: indicate the main conclusions or interpretations.
2. At the end of the Introduction section, present the structure of the article.
3. The presentation of sociodemographic data of the sample can be presented in a table that allows a better view of the results.
4. The conclusions section should be significantly improved. Present the main results obtained. I recommend that the conclusions section clearly presents the results obtained from the research, the theoretical, economic/managerial implications, and the limits of the research.
Author Response
Dear reviewer 2
Thank you very much for these comments and for the constructive feedback that allowed us to improve the article. To answer the questions raised, the abstract has been changed to clarify the “question addressed in a broad context; Methods: briefly describe the main methods or treatments applied; Results: summarize the article’s main findings; Conclusions: indicate the main conclusions or interpretations”.
The introduction section was also changed in order to present the structure of the article.
In the sample section, a table was introduced (Table 1).
In relation to the last point, we are also grateful for this suggestion however we consider the discussion section detailed the results obtained from the research, managerial implications, and the limits of the research in this sense the conclusion section is shorter so as not to repeat the ideas set out in the previous section.
Thank you very much once again for your suggestions.
This manuscript is a resubmission of an earlier submission. The following is a list of the peer review reports and author responses from that submission.
Round 1
Reviewer 1 Report
I would like to apologize for my English because I know it is not perfect.
First, I would like to start by thanking the authors for the effort made in the work.
First of all, I would like to know why you are doing two studies of two very different sectors for your research because they could be two different articles.
The abstract, it should contain the relevance of the chosen topic, sample, methodology, and main findings obtained.
Regarding the introduction, it should follow the same structure as the abstract but a little more extended, including at the end a paragraph about the structure of the article. It cannot contain the research questions there.
The literature review, since it is a work that deals specifically with employees, must relate the literature to them. It cannot be general literature, it must contain previous studies related to the article. In addition, at the end of each section, it should indicate which is the hypothesis to be tested. In the work to be done a regression analysis cannot be established as research questions.
Regarding the empirical research
There is no technical research sheet where it is clearly shown which is the universe, the sample, the moment of time in which it is carried out, the level of confidence, etc.
They do not indicate whether the hypotheses are accepted or not
The presentation of sociodemographic data of the sample can be presented in a table that allows a better view of the results.
The statistical study is not complete; in addition to presenting the descriptive data of the variables and the regression, it would be interesting to present the correlation matrix.
It is preferable to go deeper into a single study than to present two half studies.
Reviewer 2 Report
The research tackles an interesting topic with the question what CSR practices do increase employee identification and whether there are differences between sectors.
Unfortunately, the approach looks rather mechanical applying analytical statistics to data whose relevance and representativeness remain in doubt in a number of ways.
Why the health sector has been chosen is not really explained, nor why the comparison with hotels provides useful insights. The study was carried out in Portugal (the reader learns that halfway through the paper in passing). Working conditions are only discussed for hotels, but not for Portugal. Instead the reference come from very different contexts (fast-food in Quebec) and look arbitrary in their assembly. A look at Portuguese statistics and a comparision with other European or Global data would be more meaningful.
The research as presented does not consider the practices of the organizations where the employees studied work. Could the results be due to the conditions in the sector or to the level of implementation of CSR? As it stands, it is unclear what the responses measure. Have the practices been assessed?
Bias is not adequately discussed in a number of ways. This starts with the selection of the hotels and the one hospital included. Why these sectors? Why these hotels and this hospital? Presumably they have CSR policies. This is mentioned only for the hospital. A comparision between sectors based on a single hotel is rather courageous.
The employees who responded to the survey are very unevenly spread in the case of hotels. How representative are they? Is there a sampling bias? Is there a difference between the groups?
What is more, the differences between the sectors regarding environmental impacts or working conditions and labar practices are not discussed. In the case of the hospitals, there is no discussion of labor practices at all.
While this may be disappointing after all your effort, but you would have to extend your research to arrive at meaningful and valid conclusions.
You need a better justficiaton for why the sectors have been chosen and how this comparison is relevant to the questions you posed.
For hospitals, you need a larger sample.
If that is impossible, can you find a non CSR but otherwise similar hospital? That would shed light on the question in the health sector at least in an exploratory fashion and may be more straightforward than a comparision with hotels where explanations for the findings and differences between the sectors are largely absent.
A range of biases needs to be treated more explicitly.
Trust you find the comments helpful, though they may not be what you hoped for.
Reviewer 3 Report
I am pleased to have the opportunity to review this research paper entitled “Corporate Social Responsibility: The impact of employees’ perceptions on Organizational Identification” - Admsci-1837042. However, the topic of this research study is interesting and fits within the journal scope. The authors should apply the comments below to increase the quality of research.
1. At the end of the Introduction section, it would be good to present: i) the purpose of the research; ii) the novelty of the research; iii) the usefulness of research for various stakeholders; iv) the paper research gap and originality; v) present the paper structure.
2. The paragraph on lines 202-206 can be moved to the end of the introduction (“In this context, this study aims to examine how employees perceive CSR practices and how these practices (centered on the environment, customers, or employees) influence their identification with the organization. The following section presents two empirical studies carried out to gather empirical evidence to address these research questions.”)
3. The Literature review section must be significantly improved, considering the numerous studies carried out at the international level regarding Corporate social responsibility. Corporate social responsibility has been a highly researched subject in recent years. Thus, I believe the analysis of specialized literature in this field must be based on articles published in the last 2-3 years. A more detailed analysis of the research in the field will help to reflect the gap in the literature and the actual contribution of this research.
4. Authors must justify why they chose two different sectors of activity for the case studies: the hotel sector and the medical (private hospital) sector.
What is the connection between the two sectors of activity?
Comparing the medical sector between public and private hospitals was much more appropriate.
This paragraph does not suggest the novelty of your research, nor does it support the choice of the two sectors of activity, which are so different and with opposite characteristics. („We believe to have contributed to the advancement of knowledge in this field, mainly by presenting empirical evidence which allows for a comparison between the hotel sector (where many studies have been carried out) and the health sector (where there are rather few))”.
5. The conclusions section should be significantly improved. Present the main results obtained and the connection between the two analyzed sectors of activity. I recommend that the conclusions section clearly presents the results obtained from the research, the theoretical, economic/managerial implications, and the research limits.
At this point, I recommend rejecting the article and reorganizing it. I wish success to the authors in the process of reorganization and improvement of the article.